# Compositional Tailoring of Mg–2Zn–1Ca Alloy Using Manganese to Enhance Compression Response and In-Vitro Degradation

**DOI:** 10.3390/ma15030810

**Published:** 2022-01-21

**Authors:** Somasundaram Prasadh, Gururaj Parande, Manoj Gupta, Raymond Wong

**Affiliations:** 1Faculty of Dentistry, National University Center for Oral Health Singapore (NUCOH), National University of Singapore, 9 Lower Kent Ridge Road, Singapore 119083, Singapore; e0204949@u.nus.edu; 2Department of Mechanical Engineering, National University of Singapore, 9 Engineering Drive 1, Singapore 117576, Singapore; mpeguru@nus.edu.sg

**Keywords:** magnesium alloy, manganese, biodegradable implants, corrosion, mechanical strength, hank’s balanced salt solution

## Abstract

The present study investigates Mg–2Zn–1Ca/XMn alloys as biodegradable implants for orthopedic fracture fixation applications. The effect of the presence and progressive addition of manganese (X = 0.3, 0.5, and 0.7 wt.%) on the degradation, and post-corrosion compressive response were investigated. Results suggest that the addition of manganese at 0.5 wt.% improved the corrosion resistance of Mg–2Zn–1Ca alloys. The pH values stabilized for the 0.5Mn-containing alloy and displayed a lower corrosion rate when compared to other Mg–2Zn–1Ca/Mn alloys. Mg–2Zn–1Ca showed a progressive reduction in the compressive strength properties at the end of day 21 whereas Mg–2Zn–1Ca/0.3Mn and Mg–2Zn–1Ca/0.5Mn samples showed a decrease until day 14 and stabilized around the same strength range after day 21. The ability of Mg–2Zn–1Ca/0.5Mn alloy to develop a network of protective hydroxide and phosphate layers has resulted in the corrosion control of the alloy. Mg–2Zn–1Ca/0.7Mn displays segregation of Mn particles at the grain boundaries resulting in decreased corrosion protection. The mechanism behind the corrosion protection of Mg–2Zn–1Ca alloys was discussed.

## 1. Introduction

In the last decade, magnesium and its alloys have been extensively researched for potential applications in the field of orthopedics, craniofacial and cardiovascular applications [1,2,3,4,5,6]. Magnesium and its alloys have gained great attention as a biodegradable implant material because of its low density, high mechanical strength, Young’s modulus (40–45 GPa) matching close to the human bone (3–20 GPa), and excellent biocompatibility [7,8,9]. These advantages of magnesium alloys have made them an exciting alternative in comparison to non-biodegradable metals like titanium alloys, cobalt–chromium alloys, and stainless steel for orthopedic and craniofacial applications [10,11,12]. The drawback of currently used bio-metals/alloys is the requirement of revision surgery to remove implants from the human body. On the other hand, magnesium is bioresorbable and can replace these non-bioresorbable materials in temporary fracture fixations [13,14,15]. However, the relatively increased degradation rate of magnesium in the physiological body solutions makes its applicability as a temporary implant material challenging [16,17,18]. Magnesium releases hydrogen gas (H_2_) when it comes in contact with the physiological solution leading to inflammatory reactions at the implanted site and further corrosion of the material reduces the mechanical integrity of the implanted material to withstand the applied load [5,19,20]. Therefore, to overcome these drawbacks, magnesium should be combined with judiciously chosen alloying elements capable to reduce the degradation rate in the physiological medium [21,22,23,24].

Alloying of various biocompatible elements into magnesium has been shown to improve the biocorrosion resistance [25,26,27]. From a biosafety perspective, the addition of endogenous alloying elements is a viable option that can, in addition, assist in circumventing the issue of host rejection [28,29,30].

Zinc (Zn) and calcium (Ca) are common essential elements required by the human body to perform various biochemical reactions and can be used to improve the overall properties of magnesium [31]. The addition of Zn and Ca have been shown to reduce the degradation rate and enhance the corrosion resistance of magnesium [32,33,34]. Zn has been shown to increase the electrode potential of magnesium in simulated body fluid and reduce the corrosion rate of Mg-6Zn binary alloys when compared with pure magnesium [35]. Yan et al. [36] stated that the progressive addition of 1–5 wt.% of Zn reduces the degradation rate of pure magnesium in simulated body fluid. Calcium exists in the form of hydroxyapatite (HA) in the human body. Alloying of calcium improves the strength of magnesium and modifies the microstructure thereby reducing corrosion [37]. Higher Ca content in Mg alloys leads to the formation of a large volume of Mg_2_Ca secondary phase, which reduces the corrosion resistance of Mg alloys [38,39]. Mohammed et al. [40] reported that the addition of 0.8 wt.% Ca to Mg leads to a reduction in the corrosion rate compared to pure Mg [40]. Liu et al. 2015 [41] evaluated the biosafety and corrosion response of Mg-30% Ca alloys. Black powder precipitate particles were released as degradation products which were made of the outer shell of Mg(OH)_2_, MgO, and Mg/Ca mixture, and the addition of Ca reduced the corrosion rate [41]. In order to retain the strength enhancements and improve the corrosion resistance, alloying of an Mg–Zn–Ca alloy system with a suitable quaternary element can be considered.

Manganese (Mn) is an essential trace element for physiological processes, and it is a necessary element for the immune system and a variety of enzymes [42,43]. Mn addition reduces the grain size thereby enhancing grain refinement [26,27]. The addition of Mn forms alpha Mn precipitates which hinders the growth of grains during the process of extrusion. Mn has also proven to be a promising element in reducing the corrosion rate of magnesium alloys [13,44]. Addition of 0.5 wt.% of Mn reduced the corrosion rate of Mg–2Ca–0.5Mn–2Zn alloys [28]. Mn forms oxide films on the corroding surface and acts as a protective barrier preventing the penetration of chloride ions [45]. Gu et al. [46] added 1 wt.% of Mn to pure Mg and found that the Mn addition reduced the corrosion rate and increased the mechanical strength of pure Mg [46]. However, the addition of Ca >1 wt.% resulted in hot tearing of the surface thereby leading to subsequent processing irregularities. Compositional control of Ca to ≤1 wt.% in conjunction with microalloying with biocompatible Mn becomes crucial to optimizing the properties of the Mg alloy. Furthermore, there has not been any past effort in literature to study the post-corrosion compression strengths of manganese containing magnesium alloys. As the material is a suitable replacement for permanent titanium fracture fixation plates and screws, strength retention through corrosion control as a result of compositional tailoring, in the absence of any heat treatment or surface modification techniques such as coating or shot peening etc. forms the key motivation of this work.

Accordingly, in this study manganese in various amounts (0.3, 0.5, and 0.7 wt.%) was added to Mg–2Zn–1Ca ternary alloy and tested for in vitro corrosion rate and pH variation by immersion corrosion/weight loss methods in Hanks balanced salt solution (HBSS) solution. Post-corrosion compression studies of ultimate compressive strength (UCS) were performed to quantify the strength retention ability of the developed Mn-based alloys. There highlights of the study are,Mg–2Zn–1Ca/XMn (X = 0.3, 0.5, and 0.7) alloys were developed using disintegrated melt deposition followed by hot extrusion.The corrosion resistance of Mg–2Zn–1Ca alloys was improved with the addition of 0.5 wt.% Mn element.The enhanced corrosion resistance is due to improved corrosion product film.Post-corrosion compressive strength of Mg–2Zn–1Ca/0.3Mn and Mg-2Zn1Ca/0.5Mn stabilized after day 21.

## 2. Materials and Methods

### 2.1. Materials

The details of metallic elements (morphology and source) used in the present study to fabricate alloys are listed in Table 1. The chemical composition of each alloy following processing was analysed by inductively coupled plasma optical emission spectroscopy (ICP-OES), the results of which are shown in Table 2. Common elements in the Periodic Table were identified and quantified up to 90%. This was supported by instruments such as the Thermo Fisher Scientific Flash Smart CHNS Elemental Analyser (CE Elantech, Lakewood, NJ, USA) and a Perkin Elmer Avio 500 Inductively Coupled Plasma-Optical Emission Spectrometer (ICP-OES) (CE Elantech, Lakewood, NJ, USA). Sample preparation was undertaken using acid digestion on the hotplate and in a microwave digester. Samples were digested with HNO_3_ in the microwave at 240 °C for 15 min and topped up to 10 mL with H_2_O. A clear solution was observed before analysis.

### 2.2. Processing

#### 2.2.1. Primary Processing Method

Disintegrated melt deposition (DMD) is a unique cost-saving processing methodology that combines the advantages of conventional casting and spray processing to produce bulk composite material using lower disintegrating gas jet velocities. Mg–2Zn–1Ca/XMn was synthesized using the DMD process [47]. The process involves heating the Mg chips/turnings/ingots with the alloying element arranged in a unique sandwich form up to a superheated temperature of 750 °C in a graphite crucible. The alloy melt is stirred at 465 rpm for 5 min to facilitate the uniform distribution of the components and to circumvent gravity segregation. This melt was bottom-poured subsequently into a metallic substrate after the disintegration from two jets of argon gas oriented normal to the melt stream to obtain 40 mm diameter ingots [47].

#### 2.2.2. Secondary Processing Method

The as-cast billet was soaked at 400 °C for 1 h in a constant temperature furnace before extrusion at 350 °C at an extrusion ratio of 20.25:1 using a 150-ton hydraulic press. Colloidal graphite was used as the lubricant. Cylindrical rods of 8 mm diameter were obtained. Detailed information can be found in the following reference [48]. The samples from the extruded rods were characterized as per ASTM standards.

### 2.3. Characterization

For investigating the microstructure a JEOL JSM-5800 LV Scanning Electron Microscope (SEM, Kyoto, Japan) coupled with energy-dispersive spectroscopy (EDS, Kyoto, Japan)) was used. Hanks balanced salt solution (HBSS) was used to test the immersion corrosion of the alloys. The setup was immersed in a water bath maintained at 37 °C to simulate the temperature of the human body. The sample dimensions of 5 mm diameter and 5 mm thickness were used. The solution to sample ratio was maintained at 20 mL:1 cm^2^. Weight loss and pH measurements were measured every 24 h. The detailed protocol ASTM G31-72 can be referred to from our previous studies [49]. Quasi-static compression tests were conducted as per ASTM E9-89a to investigate the post-corrosion compressive properties using an 810 MTS (Material Testing System) under the strain rate of 8.3 × 10^−5^ s^−1^.

## 3. Results

### 3.1. Immersion Studies—Corrosion Rate and pH

The corrosion rate and pH measurements are shown in Figure 1 and Table 3. The pH values remained in the range of ~9.4–9.8 (Figure 1A). The corrosion rates at the end of day 1 for Mg–2Zn–1Ca, Mg–2Zn–1Ca/0.3Mn and Mg–2Zn–1Ca/0.7Mn were ~0.19 mm/y, ~0.43 mm/y, ~0.21 mm/y, respectively. However, the Mg–2Zn–1Ca/0.5Mn displayed a lower corrosion rate at ~0.07 mm/y. Furthermore, at the end of day 28, the corrosion rates were gradual and steady for 0.5Mn samples reaching ~0.25 mm/y. However, the corrosion rates for the samples Mg–2Zn–1Ca, Mg–2Zn–1Ca/0.3Mn reached ~0.87 mm/y and ~1.82 mm/y, respectively (Figure 1B). Mg–2Zn–1Ca/0.7Mn groups showed the maximum corrosion rate compared to all the groups with the corrosion rate reaching up to ~2.15 mm/y at the end of day 28 (Figure 1B and Table 3).

The post-corrosion compression behavior was studied, and the results are shown in Figure 1C and Table 4. Mg–2Zn–1Ca showed a progressive reduction in the compressive strength properties at the end of day 7, day 14, and day 21. Mg–2Zn–1Ca/0.3Mn and Mg–2Zn–1Ca/0.5Mn samples showed a decrease till day 14 and stabilized around the same strength range. Mg–2Zn–1Ca/0.7Mn sample decreased in strength until day 14 and dissolved completely at the end of day 21.

### 3.2. Microstructure Characterisation—Grain Size

Figure 2 shows the scanning electron microscopic images of grain morphology. The grain size of Mg–2Zn–1Ca ternary alloys were 23 ± 7 μm whereas the grain size of 0.3Mn, 0.5Mn and 0.7Mn addition to Mg–2Zn–1Ca alloys were 18 ± 8 μm, 16 ± 4 μm and 18 ± 7 μm respectively (Table 4). The grain size of the Mg–2Zn–1Ca alloy is refined by ~22%, 30% and 22% respectively with the progressive addition of 0.3, 0.5 and 0.7 wt.%. The microstructure analysis reveals the effect of extrusion to break down the secondary phases and distribute uniformly on the surface of the samples. The refinement in grain size of the samples is known to improve the corrosion resistance in Mg alloys. The finer grain size of Mg alloys improves the passivity of the surface thereby increasing the corrosion resistance [50].

### 3.3. Post-Corrosion Scanning Electron Microscopy (SEM)

Figure 3 and Figure 4 shows post corrosion scanning electron microscopic images. In the present study, pits of different sizes on the surface of the samples were observed. Among the alloy samples, the number of pits and non-uniform corrosion observed in the 0.7Mn group was comparatively higher than the other samples. The Mg(OH)_2_ layer formation in the 0.7Mn sample was noted to be non-uniform (Figure 3). Uniform passive layer formation was observed for the groups containing 0.5Mn and 0.3Mn (Figure 3 and Figure 4). EDS mapping results of the sample reveal a high amount of Mg and O (oxide) and P (phosphorous) and this is feasible for apatite layer formation (Figure 5, Figure 6 and Figure 7).

## 4. Discussion

Figure 2 shows the microstructural distribution of the synthesized Mg alloys displays a bimodal grain distribution. It can be observed that all the alloys show three distinct features; finer grains marked by red arrows, coarser grains marked by green arrows and secondary phase particles marked by yellow arrows. The finer grains are a result of dynamic recrystallization (DRX) during the extrusion process, whereas the coarser grains are the deformed grains. This observation is in agreement with reported works on Mn-containing Mg alloys [51].

Critical corrosion resistance is crucial to retain the load-bearing strength with minimum inflammatory behavior in order to target magnesium-based alloys as a bioresorbable implant until the time of wound healing [52,53]. Mg and Mg-based materials interact with the physiological environments leading to an increase in the pH for the initial 12–24 h [54].

This leads to the release of Mg^2+^ ions into the solution due to the anodic dissolution of Mg as observed in Figure 1B. The pH values remained in the range of ~9.4–9.8 (Figure 1A). However, pH values from day 2 to day 28 for all the samples remained steady, and no substantial increase or decrease in the values was observed. The absolute pH values of the 0.3Mn, 0.5Mn were observed to be lesser than that of Mg–2Zn–1Ca alloys. This suggests the compositional and microstructural characteristics of these alloys resulted in enhanced dynamic passivation of the samples from day 2 to day 28 thereby keeping the pH increase/decrease within control limits. Table 3 shows the corrosion rates of various Mn incorporated Mg–2Zn–1Ca alloys tested for orthopedic applications. The formation of the hydroxide layer is the initial cause of an increase in the pH [55]. At high pH levels, the hydroxide layer attains thermodynamic stability [56]. This behavior can be observed in the corrosion rate trends as well. Once the layer is stable, the corrosion rate tends to decrease. However, other factors like an increase in magnesium chloride (MgCl_2_) formation by the activation of Cl^−^ ions tend to increase the degradation rates of the immersed alloys [21,55,57]. The degradation rate tends to increase in acidic conditions whereas there is a decrease in basic pH environments [58].

The corrosion rates in the present study were calculated using equation 1, where, time conversion coefficient, K = 8.76 × 10^4^, W is the change in weight pre- and post-immersion (g), A is the surface area of the sample exposed to the immersive medium (cm^2^), T is the time of immersion (h), and D is the experimental density of the material (g·cm^−3^) [2].
(1)$$\mathrm{Corrosion}\;\mathrm{rate}=\frac{K\;\times \;W}{\left(A\;\times \;T\;\times \;D\right)}$$

The primary corrosion mechanism observed in the developed alloys was pitting corrosion [59]. The presence of Mn reduced the extent of pitting as long as the randomized distribution of the secondary phases was retained while a reduction in the grain size (Figure 2) is seen as the second governing mechanism. Uniform passive layer formation was observed for the groups containing 0.5Mn and 0.3Mn, which acted as a barrier between the matrix material and the immersive medium thereby delaying the onset of corrosion [60,61].

Post corrosion SEM shows among the alloys, 0.5Mn alloy samples showed a comparatively more uniform layer formation and best corrosion resistance while 0.7Mn alloy samples showed the maximum corrosion rate (Figure 1) and (Table 3). This can be attributed to: (a) the increasing presence of clusters with the increasing amount of Mn and (b) an increase in the amount of cluster associated porosity due to the increasing presence of Mn causing segregation of the alloy [62,63].

The corrosion process in biodegradable metals like Mg-based alloys includes electrochemical reactions where oxides, hydroxides, and H_2_ gas is evolved [64]. The anodic and cathodic reduction reactions are given below:Anodic reaction: Mg → Mg^2+^ + 2e^−^(2)
Cathodic reaction: 2H_2_O + 2e^−^ → H_2_ + 2OH^−^(3)
Overall reaction: Mg + 2H_2_O → Mg^2+^ + 2OH^−^ + H_2_(4)
Mg^2+^ + 2OH^−^ → Mg(OH)_2_(5)
Chloride attack: Mg(OH)_2_ + Cl^−^ → MgCl_2_ + 2OH^−^(6)
Phosphate reaction: 10Ca^2+^ + 2OH^−^ + 6HPO_4_^2−^ → Ca_10_(PO_4_)_6_(7)
3Ca^2+^ + 2OH^−^ + 2PO_4_^3−^→ (Ca)_3_(PO_4_)_2_(8)

It can be observed from Figure 2 and Figure 4 that the grain boundary morphology correlates with the corrosion behavior of the material. The corrosion initiation occurs at the grain boundaries during the anodic dissolution of the Mg cations [65]. This is attributed to the fact that the grain boundaries are relatively more anodic in nature compared to the secondary phases and the Mg matrix [51,65]. This anodic dissolution causes the release of electron transfers to the DRX region and the secondary phases (see reaction (2)).

At the anode, Mg oxidizes to form Mg^2+^ cations and releases electrons, which are utilized at the cathode for the reduction of water. These reactions occur throughout the surface. Because of the dissimilarities in electrochemical potential among the metal matrix and intermetallic (secondary) phases and with the surface adsorbed organic molecules, galvanic couples are formed, leading to the dissolution of Mg alloys [66]. Pure Mg is susceptible to corrosion in physiological conditions as they are composed of dissolved oxygen, proteins, and electrolyte ions such as chloride and hydroxide ions; thus migration of ions takes place from the metallic surface to the surrounding medium resulting in the formation of an insoluble hydroxide layer (Mg(OH)_2_) on the Mg alloy surface (see reaction (4)). A passive layer is formed, preventing the migration of ions across the metallic surface when the metallic oxide covers the surface completely [67]. This layer is preferentially formed over the DRX grains and the secondary phases. This results in a decrease in the corrosion rate as the hydroxide layer forms a diffusion barrier between the matrix and the solution. This is attributed to the grain refinement of alloys which reduces the stresses between the hydroxide layer and the underlying uncorroded metal surface, decreasing the susceptibility to cracking, resulting in improved corrosion resistance [50]. However, the coarser grains are not completely covered by this passivation layer (Figure 4). These quasi-adherent layers then become the centre of corrosion due to their anodic nature [68]. This metal oxide layer, however, can easily break down when attacked by chloride ions, causing pitting corrosion [69]. The low-radius Cl^−^ ions from the HBSS diffuse through the layer and interact with the surface resulting in localized pits and with the secondary phases causing micro corrosion pits. The amount of Mg decreases and at the same time, the amount of O from the solution to the hydroxide layer progressively increases. Furthermore, the PO_4_^3−^, and HPO_4_^2−^ ions from the HBSS react with the free OH^−^ along with Ca^2+^ from the degraded alloys to form (Ca)_3_(PO_4_)_2_ apatite layer.

There is a rapid evolution of H_2_ gas during corrosion of Mg-based alloys due to a chloride-rich environment. This chloride-rich environment vanishes after the initial few weeks post-surgery. [70,71]. It has been reported that the release of hydrogen at the rate of 0.01 mL·cm^−2^·day^−1^ would cause negligible harm [72]. Several factors affect the corrosion, such as concentration and types of ions, protein adsorption on implant, pH, and the effect of biochemical activities around the implant [73,74,75]. In addition to the controlled degradation behavior, any potential biomaterial should also encourage in vitro bioactivity to ease the bone-resorption process [20,76].

The post-corrosion UCS, as expected, decreased due to the corrosion of the materials in Hank’s solution (Table 4). All the samples exhibited retention in ultimate strength, although at a reduced level, except for 0.7Mn alloy groups at the end of day 21 immersion when compared to uncorroded condition. The high strength retention can be attributed to the homogenous solid solution effect of Zn in Mg. The slight decrement in the UCS values in Mg-2Zn1-Ca alloy and Mg–2Zn–1Ca/0.5Mn alloy can be attributed to the corrosion effects such as loss of materials and pit formation on the surface that led to increased crack nucleation sites, as shown in Figure 3. The ultimate compressive strengths showed a reduction in the values after day 14 immersion suggesting the progressive adverse effect of pit formation on the surface (increased crack nucleation sites) leading to a compromise in the UCS (Table 5). Table 6 shows the comparison of corrosion rates of the developed alloys in the present study with commercial magnesium alloys containing the manganese element measured in simulated body fluid (SBF) and Hank’s solution.

## 5. Conclusions

Inspired by the attractive potential of Mg-based alloys in biomedical applications, in this study the influence of manganese addition on the biocorrosion properties and post-corrosion strength retention of Mg–2Zn–1Ca alloy was studied. The following conclusions were drawn:The effect of manganese addition on the bio-corrosion behavior and post corrosion strength retention was successfully established for Mg–2Zn–1Ca alloy.The results showed the lowest corrosion rate for Mg–2Zn–1Ca/0.5Mn alloy. This can be attributed to the optimized presence of secondary phases (near absence of clusters) and reduced grain size. Corrosion resistance of the alloys can be represented as: Mg–2Zn–1Ca/0.5Mn > Mg–2Zn–1Ca > Mg–2Zn–1Ca/0.3Mn > Mg–2Zn–1Ca/0.7Mn.The post-corrosion compression properties displayed superior strength retention for all the samples except Mg–2Zn–1Ca/0.7Mn. The relative order of post-corrosion strength retention of the alloys can be represented as: Mg–2Zn–1Ca/0.5Mn > Mg–2Zn–1Ca > Mg–2Zn–1Ca/0.3Mn > Mg-2Zn1-Ca/0.7Mn.The presence of a controlled amount of Mn addition to the Mg2Zn1Ca alloy up to 0.5 wt.% can be attributed to the ability of the material to develop a uniform passivating layer as a result of the formation of the hydroxide and phosphate compounds. Addition of Mn above 0.5 wt.% has resulted in the decrement in the corrosion resistance of Mg–2Zn–1Ca owing to the non-uniform segregation of Mn along the grain boundaries of the alloy.

## Figures and Tables

**Figure 1 materials-15-00810-f001:**
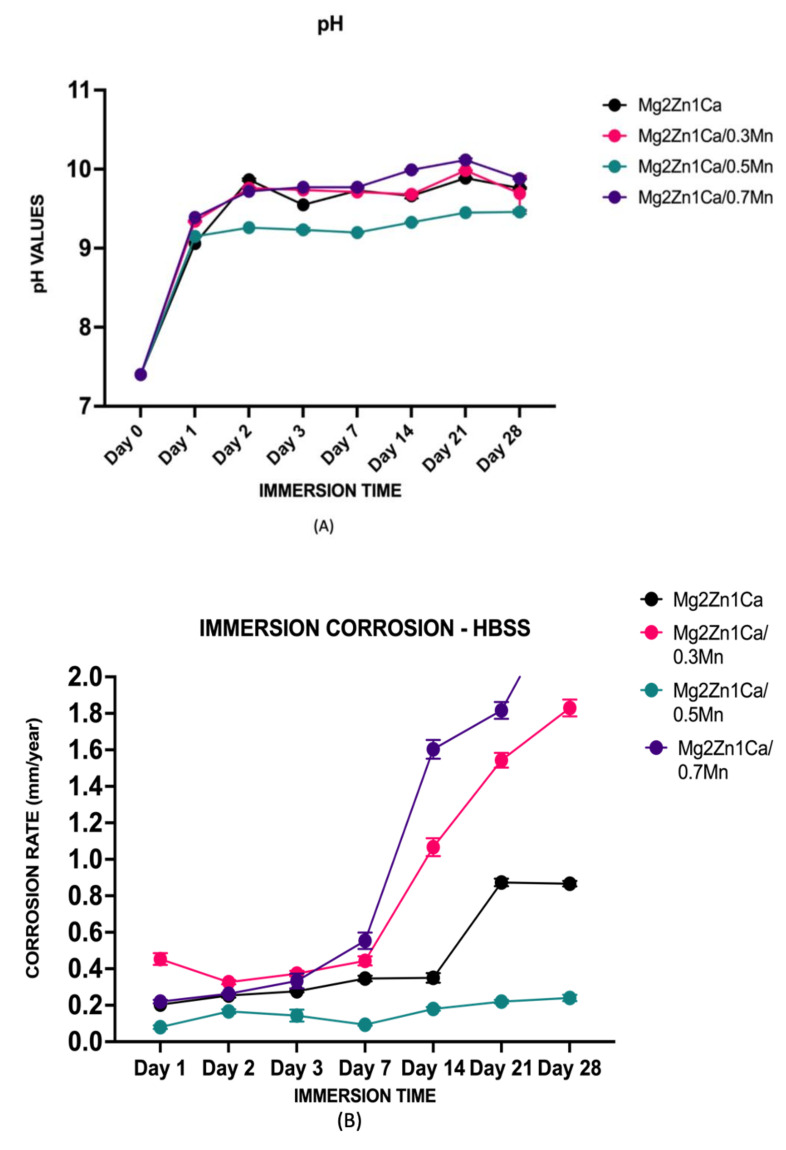
Immersion corrosion rates, pH and post corrosion compressive strength of magnesium alloys in Hanks balanced salt solution (HBSS). (**A**) pH vs. time of immersion; (**B**) corrosion rate vs. time of immersion (**C**) post-corrosion ultimate compressive strength (UCS). The loading direction was parallel to the extrusion axis direction. (* *p* < 0.05).

**Figure 2 materials-15-00810-f002:**
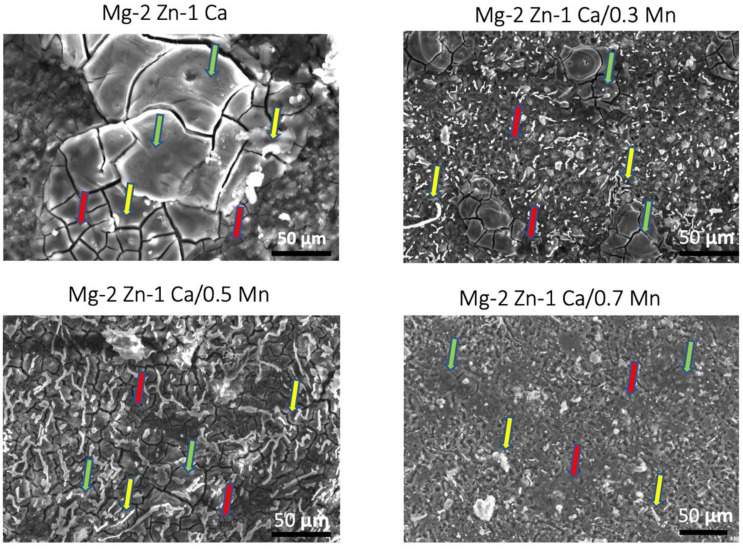
Scanning electron microscope images showing grain size and morphology of Mg–2Zn–1Ca, Mg–2Zn–1Ca/0.3Mn, Mg–2Zn–1Ca/0.5Mn and Mg–2Zn–1Ca/0.7Mn (Finer grains marked by red arrows, coarser grains marked by green arrows and secondary phase particles marked by yellow arrows).

**Figure 3 materials-15-00810-f003:**
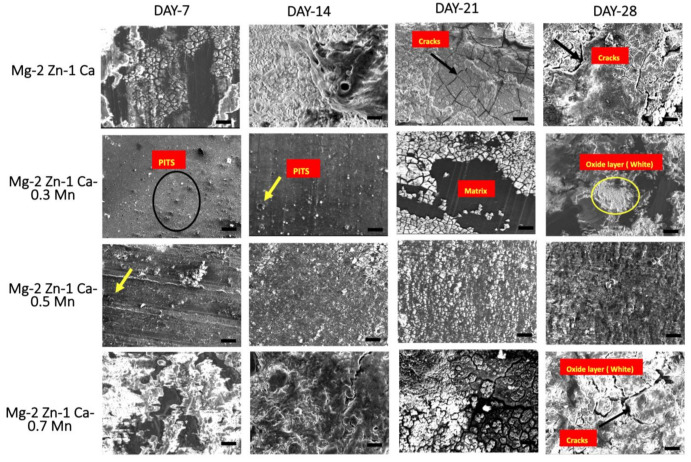
Scanning electron micrograph images of magnesium alloys after 7, 14, 21, and 28 days of immersion in HBSS solution. Scale bar 100 µm.

**Figure 4 materials-15-00810-f004:**
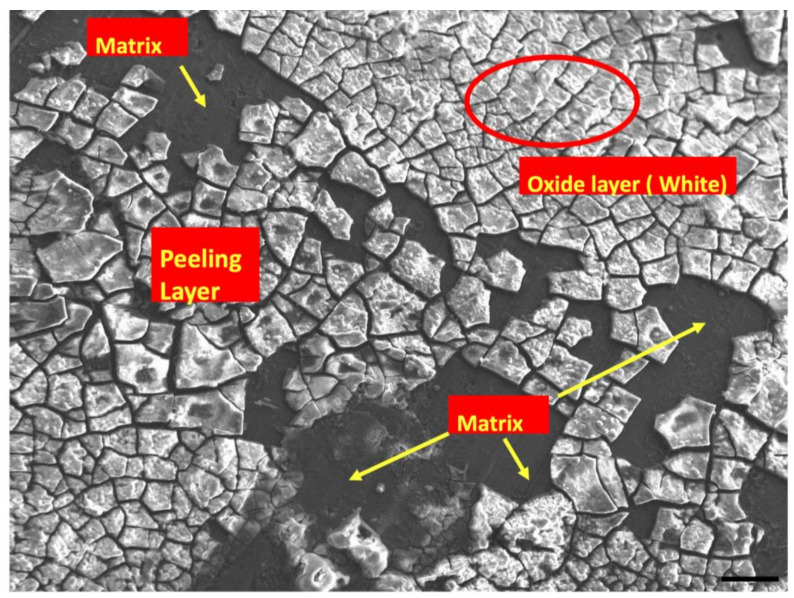
SEM analysis of Mg–2Zn–1Ca/0.5Mn alloys after 28 days of HBSS immersion. Scale bar 100 µm.

**Figure 5 materials-15-00810-f005:**
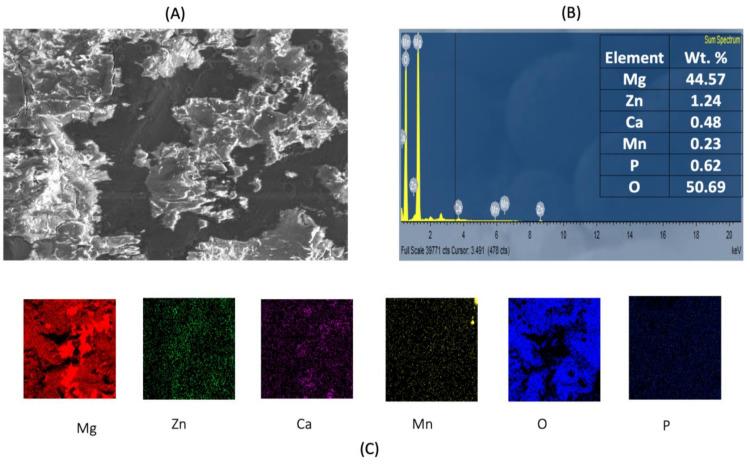
Energy-dispersive X-ray spectroscopy (EDS) analysis of Mg–2Zn–1Ca/0.3Mn alloys. (**A**) High-magnification image of the matrix; (**B**) sum spectrum and weight % values; (**C**) EDS mapping of Mg, Zn, Ca, Mn and O. Scale bar 100 µm.

**Figure 6 materials-15-00810-f006:**
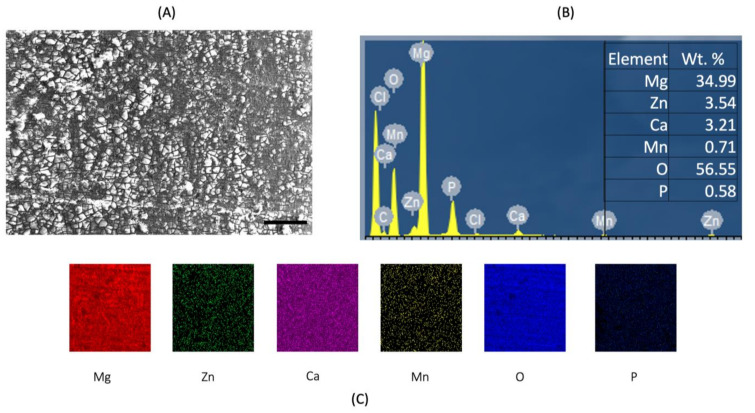
EDS analysis of Mg–2Zn–1Ca/0.5Mn alloys. (**A**) High-magnification image of the matrix; (**B**) sum spectrum and weight % values; (**C**) EDS mapping of Mg, Zn, Ca, Mn and O. Scale bar 100 µm.

**Figure 7 materials-15-00810-f007:**
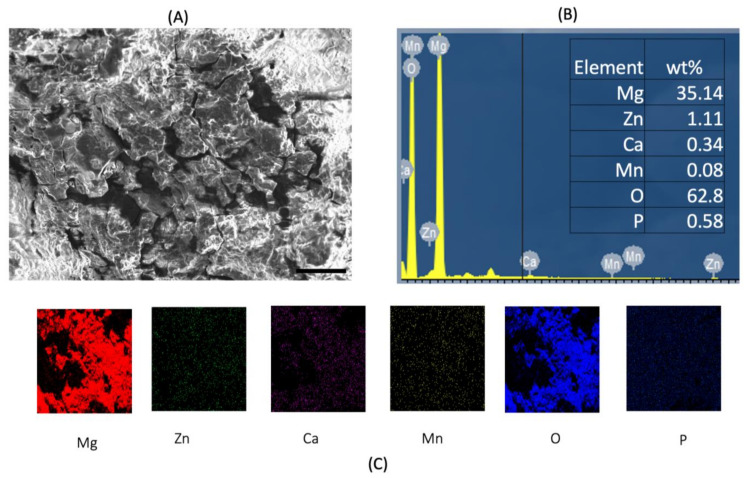
EDS analysis of Mg–2Zn–1Ca/0.7Mn alloys. (**A**) High-magnification image of the matrix; (**B**) sum spectrum and weight % values; (**C**) EDS mapping of Mg, Zn, Ca, Mn and O. Scale bar 100 µm.

**Table 1 materials-15-00810-t001:** Specification of materials utilized in the current study.

Raw Material	Size (Morphology) Purity	Supplier
Mg	Turnings	ACROS Organics, USA
	>99.9%	
	~149 µm	
Zn	(Spherical)	Alfa Aesar, USA
	99.9%	
Ca	granules	
	99.9%	Alfa Aesar, USA
	~100 µm	
Mn	(Spherical)	Alfa Aesar, USA
	99.9%	

Mg: magnesium; Zn: zinc; Ca: calcium; Mn: manganese.

**Table 2 materials-15-00810-t002:** Chemical composition of the alloy (wt.%) according to the inductively coupled plasma optical emission spectroscopy (ICP-OES).

Material	Mg	Zn	Ca	Mn
Mg–2Zn–1Ca	Bal.	1.89	0.63	0
Mg–2Zn–1Ca/0.3Mn	Bal.	1.88	0.83	0.15
Mg–2Zn–1Ca/0.5Mn	Bal.	1.98	0.94	0.49
Mg–2Zn–1Ca/0.7Mn	Bal.	1.92	0.99	0.66

Mg: magnesium; Zn: zinc; Ca: calcium; Mn: manganese.

**Table 3 materials-15-00810-t003:** Corrosion rates of alloys for day 28 immersion in HBSS.

Immersion Time (Days)	Corrosion Rate (mm. Year^−1^)			
	Mg–2Zn–1Ca	Mg–2Zn–1Ca/0.3Mn	Mg–2Zn–1Ca/0.5Mn	Mg–2Zn–1Ca/0.7Mn
Day 1	0.19	0.43	0.07	0.21
Day 2	0.26	0.32	0.16	0.26
Day 3	0.27	0.36	0.13	0.31
Day 7	0.33	0.42	0.09	0.50
Day 14	0.32	1.10	0.19	1.56
Day 21	0.89	1.58	0.22	1.79
Day 28	0.87	1.82	0.25	2.15

**Table 4 materials-15-00810-t004:** Grain size measurements for developed materials.

**Materials**	Mg–2Zn–1Ca	Mg–2Zn–1Ca /0.3Mn	Mg–2Zn–1Ca /0.5Mn	Mg–2Zn–1Ca /0.7Mn
**Grain Size (µm)**	23 ± 7	18 ± 8 (↓ 22%)	16 ± 4 (↓ 30%)	18 ± 7 (↓ 22%)

**Table 5 materials-15-00810-t005:** Comparison of pre-/post-corrosion compression properties of the magnesium alloys [47,77,78].

Material	0.2CYS (MPa)	UCS (MPa)	Fracture Strain (%)
Mg–2Zn–1Ca	148	481	18.8
Mg–2Zn–1Ca	111± 11	302	10 ± 2
(Day 14)	(↓ 25%)	(↓ 37%)	(↓ 44%)
Mg–2Zn–1Ca/0.3Mn	226	568	11.3
Mg–2Zn–1Ca/0.3Mn	155 ± 10	301 ± 14	13 ± 1
(Day 14)	(↓ 31%)	(↓ 47%)	(↑ 15%)
Mg–2Zn–1Ca-/0.5Mn	214	645	13.4
Mg–2Zn–1Ca/0.5Mn	211 ± 16	352 ± 12	11 ± 3
(Day 14)	(↓ 1%)	(↓ 45%)	(↓ 15%)
Mg–2Zn–1Ca/0.7Mn	249	565	13.2
Mg–2Zn–1Ca/0.7Mn	85 ± 12	144 ± 9	9 ± 1
(Day 14)	(↓ 66%)	(↓ 74%)	(↓ 30%)
Natural bone	130–180	-	-
Cortical bone	-	131–224	2–12
Stainless steel	-	170–310	-
Titanium	-	758–1117	29–49
alloy			
Co-Cr alloy	-	450–1000	-
AZ91D	130	300	12.4
AM50	110	312	11.5
WE43	261	420	16.3

CYS: Compressive Yield Strength; UCS: Ultimate Compressive Strength.

**Table 6 materials-15-00810-t006:** Comparison of the lowest corrosion rate of the magnesium alloys containing manganese elements using simulated body fluid (SBF) and Hank’s solution [27,79,80].

Alloys	Solution	Corrosion Rate (mm/y)
Mg–2Zn–1Ca/0.5Mn (This work)	Hank’s	0.25
Mg-4Zn-0.5Ca-0.75Mn	Hank’s	0.12
ZK60	Hank’s	0.32
MAO-coated ZK60	Hank’s	0.003
Mg-2Zn-Ca-0.5Mn	Hank’s	1.58
Mg-2Zn-Ca-0.5Mn-1.3Ce	Hank’s	1.34
Mg–2Ca–0.5Mn–2Zn	SBF	1.78
Mg–2Ca–0.5Mn–4Zn	SBF	2.21
Mg–2Ca–0.5Mn–7Zn	SBF	3.98
Mg-Zn-1.5-Ca-1.1Mn	Hank’s	1.40
6h-coated Mg-2Zn-Ca-0.5Mn-1.3Ce	Hank’s	1.29
Mg-Zn-Ca BMGC	SBF	0.26
Annealed Mg-Zn-Ca	SBF	1.20

SBF: Simulated Body Fluid.

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
