# Peer review of "Compositional Tailoring of Mg–2Zn–1Ca Alloy Using Manganese to Enhance Compression Response and In-Vitro Degradation"

_materials, 2022, doi:10.3390/ma15030810_

Round 1
Reviewer 1 Report
The paper studied the effects of Mn addition on the degradation and post corrosion compressive response of Mg-2Zn-1Ca alloys. However, the characterization of microstructure in this paper was not convincing, and the deep discussion was lacked. Hence, a major revision is needed.
(1) “Mg-2Zn-1Ca/χ-Mn” should be “Mg-2Zn-1Ca/χMn”.
(2) From a biomedical viewpoint, the advantages of Mn compared to the other alloying elements, such as Sr and Sn, should be supplemented.
(3) P. 4, the position for microstructure observation and the compressive direction of the quasi-static compression tests should be delivered.
(4) P. 6, Fig. 2 gives the information of grain size and the second phase particles, which was crucial to the mechanical properties and corrosion behavior. However, the quality of Fig. 2 was unacceptable. Please verify the sample was chemically etched using an appropriate metallographic etchant.
(5) P. 6, “The Mg(OH)2 layer formation was observed……” How was the components of the corrosion product determined.
(6) P. 8, the (A), (B) and (C) should be labeled in Fig. 5.
(7) P. 8, the effects of variation of pH values on human bodies should be mentioned. Moreover, the proper range of pH values should be given.
(8) P. 8, when calculated the corrosion rate based on weight changes, the weight was obtained with or without corrosion products?
(9) P. 10, the “thogh” should be “though”.
(10) The microstructure of Mg alloys including grain size and the second phase particles was altered after the addition of Mn element, which can strongly affect the corrosion resistance and post corrosion compressive response. However, their relationship was not discussed in the paper. Here are several references might be helpful, Corrosion Science 192 (2021) 109860, Corrosion Science 193 (2021) 109875, Materials Characterization 180 (2021) 111439.
Author Response
PFA

Reviewer 2 Report
There has been a growing interest in the development of Mg-based alloys for biomedical implants and related applications, owing to their superior mechanical properties and excellent biocompatibility. However, the degradation of Mg-based alloys in physiological body solutions has limited their widespread adoption. A number of studies over the past two decades have explored the use of Zn, Ca and Mn as alloying elements in Mg-based alloys to overcome the challenges associated with degradation & corrosion. This work presents an important advancement in this regard by addressing the corrosion rate and post-corrosion compression strength as a function of Mn-loading in Mg-Zn-Ca alloys. The results reveal that an optimized balance between the presence of secondary phases and the reduced grain size in one alloy composition results in improved corrosion resistance. The methodology and results are presented in a clear and concise manner. The manuscript is therefore suitable for publication in Materials. However, I would like the authors to address the following comments/ questions:
- HBSS (Hanks balanced salt solution) is not defined when described fort he first time in the introduction (Line 108)
- In Table 2, why is this sample designated as "Mg-2Zn-1Ca/0.3Mn" when there is 0.15 wt.% of Mn? Should this be "0.2Mn" instead?
- In Section 2.2.1, please provide a reference that describes the DMD technique.
- Why was 400C used as the annealing temperature? (I presume this is to homogenize the as-cast microstructure). Was a specific phase targeted with this secondary processing method?
- Was the purpose of extrusion primarily to uniformly distribute secondary phases? How come the extrusion process did not yield a texture (preferential grain orientation or elongated grains) to the microstructure?
- Was a special fixture used to prevent sample buckling during the quasi-static compression test?
- How was grain size determined (i.e., by image processing of SEM micrographs)?
- What are the bright/ white features in the SEM micrographs? Are these impurity inclusions, oxide layer or secondary phases?
- The SEM images in Figure 2 for Mg-2 Zn-1 Ca, 0.3Mn and 0.5 Mn show the appearance of cracks in the microstructure. Could the authors discuss the origin of this? It seems odd that cracks are seen even in the microstructure of pre-corroded samples.
- What is the origin of the presence of phosphorous in the corrosion layer? Is it from the HBSS solution? Please clarify.
- Why was hydrogen from the magnesium hydroxide layer picked up in the EDS scan?
- Acronym UCS (ultimate compressive strength) is not defined previously.
Author Response
PFA

Reviewer 3 Report
Authors performed the studies on “Compositional Tailoring of Mg-2Zn-1Ca alloy Using Manganese to Enhance Compression Response and Invitro Degradation”. The article provides good information. The authors should consider the following comments to revise the manuscript.
- Authors can also include the graphical representation of the experimental procedure stages for better understanding to the readers
- Line No: 140, authors can include the ASTM standard numbers
- Authors can include the photo of the sample before and after corrosion study at 28 days
- Line no 182, authors mentioned the microstructure is nearly equiaxed, author may consider rewriting equiaxed or the structure name.
- Authors can include the structures of 3 and 0.7 Mn addition.
- Figure 3, the SEM image magnification is not uniform in different samples.
- The authors have not discussed the microstructural observation in the discussion section.
- Authors can include possible XRD analysis for the phase formation analysis
Author Response
PFA

Round 2
Reviewer 1 Report
The paper can be accepted as it is.